A new scenario applying traffic flow analogy to poleward expansion of auroras

Osuke Saka

Office Geophysik, Ogoori, Japan

Abstract

Transient westward electric fields from the magnetosphere generate equatorward plasma drifts of the order of kilometers per second in the auroral ionosphere. This flow channel extends in north-south directions and is produced in the initial pulse of Pi2 pulsations associated with the field line dipolarization. Drifts in the ionosphere of the order of kilometers per second that accumulated plasmas at the low latitude end of the flow channel are of such large degree that possible vertical transport effects (including precipitation) along the field lines may be ignored. In this condition, we suggest that plasma compression in the ionosphere initiated the dynamic ionosphere. The dynamic ionosphere includes a nonlinear evolution of the compressed ionospheric plasmas, generation of field-aligned currents to satisfy the quasi-neutrality of the ionosphere, and parallel potentials associated with the excitation of an ion acoustic wave. We will study how the dynamic ionosphere created auroral expansion.

1. Introduction

"Auroras and solar corona observed at the solar eclipse are optical phenomena unique in space physics. With enough knowledge about the underlying physical processes, once auroras have been captured by a highly sensitive imager, they provide an unexpected wealth of information about plasma environment of the Earth" [Oguti, 2010]. Plasma drifts in the ionosphere observed by the balloon-measured electric fields [Kelley et al., 1971], by the Ba releases [Haerendel, 1972] and by radar

observations [Nielsen and Greenwald, 1978] that did not match the expanding trajectories of auroras were an example. They would have been observed in all-sky images as violent motion of auroras propagating poleward [Akasofu et al., 1966] or contact breakups initiated at the nearest approach to the hydrogen arc [Oguti, 1973]. To account for the differences in propagation directions, it was suggested that the primary sources of auroral particles are in the magnetospheric plasmas and they developed poleward in terms of propagation of rarefaction wave in the tail [Chao et al., 1971; Liu et al., 2012], tailward regression/braking of the fast earthward flows referred to as BBFs [Shiokawa et al., 1997; Haerendel, 2015], and onset instability of inner plasma sheet pressure [Nishimura et al., 2010]. The above explanations were based on the observations that substorm expansion was initiated and amplified at the substorm onset by the BBFs arriving at the inner boundary of plasma sheet from the tail [Kepko et al., 2004; Angelopoulos et al., 2008; Machida et al., 2009].

We show that the electric fields in the dipolarization front (DF) that are embedded in the leading edge of BBFs [Runov et al., 2011] triggered the ballooning instability of the stretched flux tubes in the inner magnetosphere. As a result, the stretched field lines returned to the dipole-like configurations by producing the convection surge. Westward electric fields associated with the convection surge were transmitted into the auroral ionosphere and yield the compressibility in auroral ionosphere. The compressibility initiates the dynamic ionosphere and leads to an alternative scenario of the poleward expansion of auroras that we discuss in this paper. In this paper, field line reconfiguration and associated auroral breakups at dipolarization onset will be summarized in section 2. In section 3, we will show that the auroral ionosphere becomes compressive transiently during dipolarization. Section 4 will discuss generation of an ion acoustic wave for creating parallel potentials in the topside ionosphere. This contributes to the outflow. Poleward expansion of discrete auroras will be discussed in section 5 in terms of a nonlinear evolution of the accumulated plasmas in the ionosphere. In the final section (section 6), we summarize our results and apply "the dynamic ionosphere" to the nonconjugate auroras.

2. Auroras and field line reconfiguration associated with geomagnetic Pi2 pulsation

Poleward expansion of auroras arising out of the onset arc was observed in the initial pulse of Pi2 pulsations [Saka et al., 2012]. Statistical study of field line inclinations at geosynchronous orbit for the intervals from 120-min prior to the Pi2 onset (T-120) to 60-min after the onset (T+60) is presented in Figure 1 (reproduced from [Saka et al., 2010]). The inclination is measured positive northward from the D-V plane of the HVD coordinates. H is positive northward parallel to dipole axis, V is radial outward, and D is dipole east. It shows that field line inclination at geosynchronous orbit (Goes5/6 at $285^{\circ}/252^{\circ}$ in geographic coordinates) decreased continuously in the growth phase and attained minimum inclination angles, $33.6^{\circ}/49.4^{\circ}$, 2-min before the initial peak of Pi2 amplitudes. These inclination angles are smaller than $57.5^{\circ}/63.8^{\circ}$ estimated by the IGRF (International Geomagnetic Reference Field) model but rather fit the T89 model [Tsyganenko, 1989] for Kp=4 ($34.2^{\circ}/45.0^{\circ}$). These field lines at the geosynchronous altitudes can be mapped to the auroral ionosphere at $63.4^{\circ}$ N/$62.7^{\circ}$N in geomagnetic coordinates by T89 for Kp=4. Following the Pi2 onset, field line inclination turned to increase in a step-like manner at Goes5 while at Goes6, which is closer to the equatorial plane than Goes5, transient dipolarization pulses were observed. From these observations, we postulate that transient electric fields in DF triggered the ballooning instability of the stretched flux tubes at the arrival of BBFs. As a result, field lines turned back to dipole-like configurations by producing the convection surge. The convection surge may be observed by the geosynchronous satellites as the convection enhancement of the plasma sheet electrons due to local breakdown of the last open trajectories of plasma sheet electrons [Thomsen et al., 2002]. The surge occurred in all-sky image coincident with the onset of bead-like rippling that leads to the breakups at the equatorward latitudes [Saka et al., 2014]. In the subsequent Pi2 pulses, an auroral surge was observed in all-sky images between $66^{\circ}$N to $74^{\circ}$N in geomagnetic latitudes, referred to as Poleward Boundary Aurora Surge (PBAS) [Saka et al., 2012]. They propagated eastward or westward at the poleward boundary of the auroral zone and were interpreted as an auroral manifestation of flow bifurcation of BBFs. In this onset scenario, the field line dipolarization finished in the initial pulse of the Pi2 pulsations and increased field line inclination in a step-like manner or generating dipolarization pulses. The

convection surge occurred once in the initial pulse of BBFs (DF) but was not repeated in the following pulses in the BBF train. This correlation suggests that auroral breakup may not repeat in the Pi2 wave packet but occurs at its initial pulse.

3. Horizontal plasma flows in the ionosphere

We assume that westward electric fields associated with the convection surge were transmitted along the field lines to the auroral ionosphere by the guided poloidal mode [Radoski, 1967]. The electric fields would be amplified during the projection into the ionosphere over 100 mV/m and created an equatorward flow through $E \times B$ drift of the order of kilometers per second in the auroral ionosphere. The flows would be confined in a flow channel expanding north-south in the midnight sector. The low-latitude end of the flow channel was at the latitudes of the onset arc. The high-latitude end may not expand beyond the poleward boundary of auroral zone. Longitudinal width of the flow channel may form a streamer [e.g., Nishimura et al, 2010] and develops after the breakups in about 1 to 2 hours of local time (~1000 km along $65^{\circ}$ N) corresponding to horizontal scale size of plasma flow vortices associated with Pi2 [Saka et al., 2014].

In the flow channel, drift across the magnetic fields for the *j-th* species ( $\mathbf{U}_{j\perp}$ ) can be written in the F region as [Kelley, 1989],

$$\mathbf{U}_{j\perp} = \frac{1}{B}[\mathbf{E} - \frac{k_B T_j}{q_j}\frac{\nabla n}{n}] \times \hat{\mathbf{B}} . \qquad (1)$$

Here, $\mathbf{E}$ denote westward electric fields in the flow channel and $\hat{\mathbf{B}}$ denotes a unit vector of the magnetic fields $B$ , downward in the auroral ionosphere. Symbols $k_B$, $T_j$, $q_j$, and $n$ are the Boltzmann constant, temperature of the *j-th* species, charge of the *j-th* species, and density of electrons (ions), respectively. The electric field of the order of 100 mV/m exceeded the diffusion (second term) by three orders of magnitudes in low temperature ionosphere. The $E \times B$ drift predominated in the F region and diffusion term may be ignored. In the E region, drift trajectories may be written [Kelley, 1989] for electrons by,

$$\mathbf{U}_{e\perp} = \frac{1}{B}[\mathbf{E} \times \hat{\mathbf{B}}] \qquad (2)$$

and for ions by,

$$\mathbf{U}_{i\perp} = b_i[\mathbf{E} + \kappa_i \mathbf{E} \times \hat{\mathbf{B}}]. \qquad (3)$$

Here, $b_i$ is mobility of ions defined as $\Omega_i/(B\nu_{in})$, $\kappa_i$ is defined as $\Omega_i/\nu_{in}$. Symbols $\Omega_i$ and $\nu_{in}$ are ion gyrofrequency and ion-neutral collision frequency, respectively. $\hat{\mathbf{B}}$ denotes a unit vector of the magnetic fields $B$. To derive equations (2) and (3), pressure gradient term (diffusion) was again ignored. In the E region ($\kappa_i = 0.1$), plasma accumulation in equatorward latitudes by the imposed westward electric fields was produced by equation (2) for electrons and the second term in (3) for ions. Electrons smoothly moved equatorward while ions stopped in the original place because of low mobilities caused by high ion-neutral collisions. However, electron accumulation in lower latitudes increased southward electric fields and simultaneously ion drifts in the first term of (3) start. If the southward electric fields grew to exceed the westward electric fields by an order of magnitudes, ion drifts in the first term of (3) and electron drifts in (2) balanced to satisfy the quasi-neutrality. This is equivalent to the generation of the Pedersen currents in the ionosphere. Thus, quasi-neutral electrostatic potential is generated in the E region, positive in poleward and negative in equatorward. The Pedersen currents would have closed to the field-aligned current (FAC), upward from the negative potential region and downward into the positive potential region to sustain the steady state electrostatic potential in the ionosphere. Plasma drifts in the ionosphere, both in E and F regions, create a cavity in the high-latitude end of the flow channel and density pileup at the low-latitude end of the flow channel. We will focus on the density accumulation in the flow channel and discuss vertical transport of these accumulated materials. The development of the cavity in the flow channel may be the subject of another paper.

4. Vertical plasma flows in the ionosphere

A transient compression of the ionospheric plasmas at the low-latitude edge of flow channel

would excite the ion acoustic wave in the ionosphere travelling along the field lines upward and downward directions from the density peak of the F region. Figure 2 shows altitude distribution of the pre-onset density profile of electrons (black) and doubled density profile caused by the accumulation in red. Electron density profile in black was plotted using sunspot maximum condition in nightside given in Prince and Bostic (1964). The travelling ion acoustic waves, upward and downward, are denoted by vertical arrows. Ion acoustic wave propagating downward may be eventually absorbed in the neutrals, while the upward wave may propagate along the field lines further upward. We will focus only on the upward travelling ion acoustic wave. Electron motions produced the parallel electric fields in accordance with the Boltzmann relation [Chen, 1974],

$$E_{//} = -\frac{k_B T_e}{q} \frac{\nabla_{//} n_e}{n_e}. \tag{4}$$

Here, $k_B$ is Boltzmann constant, $q$ is electron charge, $T_e$ is electron temperature, and $n_e$ is electron density ($n_e = n_i$). Equation (4) gives electric field strengths of the order of $0.4\,\mu V/m$ and $2.0\,\mu V/m$ for $T_e = 1000K$ and $T_e = 5000K$, respectively, when the e-folding distance of density dropout along the filed lines was 200 km. For ions, steady-state motions exist in the ionosphere in the altitudes where ion-neutral collision frequencies exceed ion acoustic wave frequencies. In that case, parallel motions can be written as [Kelley, 1989],

$$V_{i//} = b_i E_{//} - D_i \frac{\nabla_{//} n}{n} - \frac{g}{v_{in}}. \tag{5}$$

Here, $b_i$ and $D_i$ denote mobility and diffusion coefficient of ions defined by $\dfrac{q_i}{M_i v_{in}}$ and $\dfrac{k_B T_i}{M_i v_{in}}$, respectively. Symbols, $M_i$, $q_i$, $v_{in}$, and $g$ are ion mass, electric charge of ions, ion-neutral collision frequency, and gravity, respectively. Ion-neutral collision frequencies from 400 km to 1000 km in altitudes were plotted in Figure 3 using nighttime sunspot maximum condition in Prince and Bostick (1964). Frequencies of ion acoustic wave were calculated by substituting wavelength of ion acoustic wave into the dispersion relation. The wavelength was assumed to be

identical to initial accumulation distance along the field lines. We chose two cases of 1000 km and 4000 km. Phase velocity of the ion acoustic wave of the order of 1600 m/s for the electron temperatures of 5000K yields the wave frequencies of $1.6 \times 10^{-3} s^{-1}$ for the wavelength of 1000 km and $4.0 \times 10^{-4} s^{-1}$ for 4000 km. These frequencies were overlaid in Figure 3. Steady-state ion motions can be adopted up to 800 km, for a wavelength over 1000 km.

Altitude profile of steady-state ion flows were evaluated substituting 1000K for ion temperatures and the same e-folding distance in equation (4). Ions are oxygen and parallel electric fields are given by the equation (4). A snapshot of the velocity profile in altitudes from 400 km to 800 km is shown in Figure 4 for the two cases of electron temperatures, 5000K for black dots and 1000K for red dots. For the low temperature case (1000k), there occurred no ion upflow because the parallel electric fields could not overcome gravity. We suggest that electron temperatures over 2700K would be needed to excite ion upflow. When electron temperature was set to 5000K, ion velocity 15 m/s at 400 km in altitudes increased rapidly to 1369 m/s at 800 km. The altitude profile of the flow velocity in Figure 4 matched Type 2 ion outflow observed by EISCAT radar [Wahlund et al., 1992]. We conclude that the ion upflow in topside ionosphere was caused primarily by the parallel electric fields excited by the upward travelling ion acoustic wave. Below 600 km in altitudes, upflow velocity was one-to-two orders of magnitudes smaller than the equatorward drift in the flow channel. Upflow velocity became comparable to the horizontal drift over 800 km in altitudes and exceeded the phase velocity of ion acoustic wave. Parallel velocity that prevailed the ion acoustic phase velocity may excite a shock at the topside ionosphere. A part of them developed to ion acoustic double layers [Sato and Okuda, 1980; Hasegawa and Sato, 1982; Hudson et al., 1983; Ergun et al., 2002] and were observed at the altitudes of 6000 – 8000 km [Mozer et al., 1976; Temerin et al., 1982]. Those ion acoustic double layers would have produced parallel potential structures referred to as inverted-V type electric fields.

5. Nonlinear evolution of the horizontal flows

Accumulation of electrons and ions occurred at the equatorward end of the flow channel. We can

estimate a rate of accumulation by the following relation,

$$\frac{\Delta n}{\Delta t} = -n_0 \frac{\Delta U}{\Delta x}.$$ (6)

Here $n$ is plasma density, $U$ denotes drift velocity in the flow channel in $x$. Substituting

$\Delta U = 10^3 \, ms^{-1}$ and $\Delta x = 10^4 \, m$, we have $\frac{\Delta n}{\Delta t} = 10^{10} \, m^{-3} s^{-1}$ for the background density

$n_0 = 10^{11} \, m^{-3}$. This gives density pileup of the order of $\frac{\Delta n}{n_0} = 100\%$ in ten seconds. If the

equatorward drift in the flow channel is an order of $10^3$ m/s (E=100 mV/m in auroral ionosphere) and

electron production by the precipitation do not exceed the accumulation rate which was 100% of the

background density in ten seconds, both outflows and precipitation may not bring significant changes

to the flux carried by $E \times B$ drift in the flow channel. We then approximate one dimensional (along

the drift path in $x$) conservation equation in the flow channel.

$$\frac{\partial n}{\partial t} + \frac{\partial}{\partial x}(nU) = 0$$ (7)

A question arises regarding maximum accumulation of plasmas at the equatorward end of the flow

channel. One possible mechanism to suppress accumulation may be associated with the ionospheric

screening that decreased the amplitudes of penetrated (total) westward electric fields by the increasing

ionospheric conductivities. In a two-dimensional ionosphere with uniform height-integrated

conductivity, total electric fields $E$ given by a sum of the incident ($E_i$) and reflected westward electric

fields may be written as $E = \left(2\Sigma_A \big/ \left(\Sigma_A + \Sigma_P\right)\right) E_i$, where $\Sigma_A$ and $\Sigma_P$ are Alfven conductance

defined by $1\big/\mu_0 V_A$ and height-integrated Pedersen conductance in the ionosphere, respectively [Kan

et al., 1982]. Symbols $\mu_0$ and $V_A$ denote magnetic permeability in vacuum and Alfven velocity,

respectively. Amplitude ratio of total electric fields to incident electric fields is a function of

conductance ratio of Pedersen and Alfven; $E\big/E_i = 2$ for a low conductivity of the ionosphere

satisfying $\Sigma_P\big/\Sigma_A << 1$, and $E\big/E_i = 0$ for a high conductivity of the ionosphere satisfying

$\Sigma_P/\Sigma_A \gg 1$. Noting that $\Sigma_P$ is proportional to the plasma density in the ionosphere, the total

electric fields monotonically decreased with increasing plasma densities caused by accumulation itself

and by the precipitations associated with the auroral activity. Another explanation may be suggested

in the polarization electric fields (eastward) produced by the accumulation itself. These electric fields

grew quickly with density accumulation and decreased the incident electric fields (westward) by the

superposition. In addition to the above scenarios, we surmise that excess accumulation of the

ionospheric plasmas may be suppressed through the term, $(\mathbf{U} \cdot \nabla)\mathbf{U}$, in the equation of motion. From

the ionospheric screening process discussed above, we tentatively assume that flow velocity $U$ is a

function of the density $n$. Then the conservation equation (7) may be written as,

$$\frac{\partial n}{\partial t} + \frac{\partial}{\partial x} Q(n) = 0 \,. \qquad (8)$$

Here, $Q(n)$ is a mass flux defined by $Q(n)=nU(n)$. This relation can be reduced to nonlinear wave

equation,

$$\frac{\partial n}{\partial t} + c(n)\frac{\partial n}{\partial x} = 0 \,. \qquad (9)$$

Here $c(n)$ is a wave propagation velocity defined by $c(n) = U(n) + nU'(n)$, $U(n)$ is a drift velocity

in the flow channel, and $U'(n)$ denotes braking/acceleration of the drift velocity by increasing and

decreasing density. The equation (9) is often referred to as propagation of "kinematic waves" to

describe traffic flow [Lighthill and Whitham, 1955]. In the following, we use dimensionless units

normalized by $U_m$, and $n_m$. Here, $U_m$ and $n_m$ denote maximum drift velocity at $n=0$ and maximum

density for complete stops of the drift, respectively. Assuming a constant braking in the flow channel,

we define $U$ by a linear function of density $n$ as $U(n)=1-n$. Noting that $Q'(n)=c(n)$, this relation is

reduced to the equation, $Q(n)=n(1-n)$, identical to the case for the traffic flow [Whitham, 1999]. Both

the $U$ and $Q$ are plotted in Figure 5A as a function of $n$. A nonlinear evolution of the density waves is

presented in Figure 5B by the characteristic curves. In the case of vehicles in traffic, the initial flows

started from $n=0$ and stopped at $n=1.0$ by the tailback of cars. For the case of the ionosphere, the

ionospheric density started from a finite density, $n=0.3$ in Figure 5B, and increased to $n=1.0$ to

terminate the flow by the full screening. The nonlinear evolution of the density profile in time is shown in Figure 5B in colors from black ($T=T_1$), red ($T=T_2$), green ($T=T_3$), blue ($T=T_4$), and to purple ($T=T_5$). After $T=T_5$, the waves propagate upstream (poleward) as a shock. The shock velocity, $V$, is given as [Whitham, 1999],

$$V = \frac{Q(n_2) - Q(n_1)}{n_2 - n_1}. \qquad (10)$$

Here, subscript 1 is for the values ahead shock and subscript 2 is for the values behind. Noting that $Q(n_2)=0$ and substituting $Q(n_1)=n_1(n_2-n_1)$, the equation (10) can be reduced to $V=-n_1$ in dimensionless unit. The propagation velocity of the shock is related to the densities ahead. For the case of $n=0.3$ in Figure 5B, shock velocity can be estimated to be $-0.3U_m$. Here, $U_m$ denotes maximum drift velocity in the ionosphere where ionospheric screening effects vanished by the condition, $\Sigma_P/\Sigma_A \ll 1$. The shock velocity may be of the order of kilometer per second, comparable but opposite to the equatorward drift in the flow channel.

6. Summary and Discussion

We proposed that the localized electric field drift introduced compressibility in the auroral ionosphere, which in turn generated field-aligned currents in the ionosphere for the quasi-neutrality, ion acoustic wave for parallel acceleration, and auroral expansions by nonlinear evolution of the ionospheric compression. We called the compressive ionosphere as dynamic ionosphere.

We apply this dynamic ionosphere to describe the asymmetry of discrete auroras in sunlit and dark hemispheres in the nightside sector (nonconjugate auroras). We suggest that imbalance of the Pedersen conductance leads to the nonconjugate auroras: Pedersen conductance in sunlit is larger than that in dark hemisphere. Larger Pedersen conductance or weaker electric fields in the sunlit ionosphere would have caused a weaker compressibility from which ion acoustic wave may not be excited or excited with only weak parallel potentials. This condition may reduce the occurrence probability of the discrete auroras and average energy of precipitating electrons in the sunlit hemisphere as exemplified in [Newell et al., 1996; Liou et al., 2001]. Weaker electric fields in the sunlit ionosphere may also

require a longer interval to accumulate enough plasmas to excite ion acoustic wave. Such an instance is described in [Sato et al., 1998] where auroral breakups in sunlit ionosphere are delayed those in the dark hemisphere.

Finally, we note that poleward expansion as described here is an auroral event occurring in the initial pulse of Pi2 pulsations. In the succeeding pulses in the Pi2 wave trains, auroras are composed of poleward surge propagating at the poleward boundary of auroral zone (PBAS) [Saka et al., 2012]. We suppose that PBASs may be directly correlated to the reconnection processes inherent in the plasma sheet. This topic will be explored in another paper.

Acknowledgements

The author would like to express his sincere thanks to all the members of Global Aurora Dynamics Campaign (GADC) [Oguti et al., 1988]. We also gratefully acknowledge STEP Polar Network (http://step-p.dyndns.org/~khay/). Geomagnetic coordinates and footprints of the satellites are available at the Data Center for Aurora in NIPR (http://polaris.nipr.ac.jp/~aurora)

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

Figure Captions

Fig. 1: Inclination angles in degrees measured positive northward from the V-D plane from 120 min prior to the Pi2 onset (T-120) and to 60 min after the Pi2 onset (T+60) reproduced from Saka et al. (2010). Magnetometer data of Goes 5/6 were represented in HVD coordinates, H is positive northward parallel to dipole axis, V is radial outward, and D is dipole east. Epoch superposition of 30 Pi2 events and mean angles calculated from them are plotted in top and in lower panels, respectively. Mean inclination angle at 2-min before the initial peak of Pi2 amplitudes (T=0) was 33.6$^{\circ}$ for G5 and 49.4 $^{\circ}$ for G6 in dipole coordinate. Dipolarization was step-like at Goes5, while at Goes6 it was composed of dipolarization pulses. The average satellite latitudes estimated by T89 model were 10.3$^{\circ}$N and 7.9 $^{\circ}$N for Goes5 and Goes6, respectively

Fig. 2: Vertical profiles from 90 km to 1000 km in altitudes of electron number density in two conditions, pre-onset in black and after accumulation in red. Nighttime sunspot maximum condition given in Prince and Bostic (1964) was used to plot pre-onset condition. Vertical arrows directing upward and downward denote travelling ion acoustic waves propagating along the field lines from the density peak of F layer.

Fig. 3: Ion-neutral collision frequency ($\nu_{in}$) in altitudes from 400 km to 1000 km calculated using nighttime sunspot maximum condition in Prince and Bostick (1964). Wave frequencies of ion acoustic wave are overlaid for two wavelength, 1000 km and 4000 km along field lines (see text).

Fig. 4: Steady-state parallel velocity in altitudes for ions (oxygen) produced by parallel electric fields $0.4 \mu V / m$ ($T_e$=1000K) in red dots and $2.0 \mu V / m$ ($T_e$=5000K) in black dots. Vertical flows in altitudes from 400 km to 800 km are shown. Flow velocity is positive upward and negative downward.

Fig. 5: (A) Normalized flux($Q$)-density($n$) curve (thin curve) and velocity($U$)-density($n$) line (thick

line) in flow channel. Vertical scale of *U-n* line is shown to the right, scale of *Q-n* curve is to the left. Dotted line at *n=0.5* indicates the critical density where *c(n)* vanishes; waves are stationary relative to the ground. Waves propagate forward/backward at a density below/above the critical density. (B) Nonlinear evolution of the density accumulation. Density increased in a step like manner from $T_1$ to $T_5$.

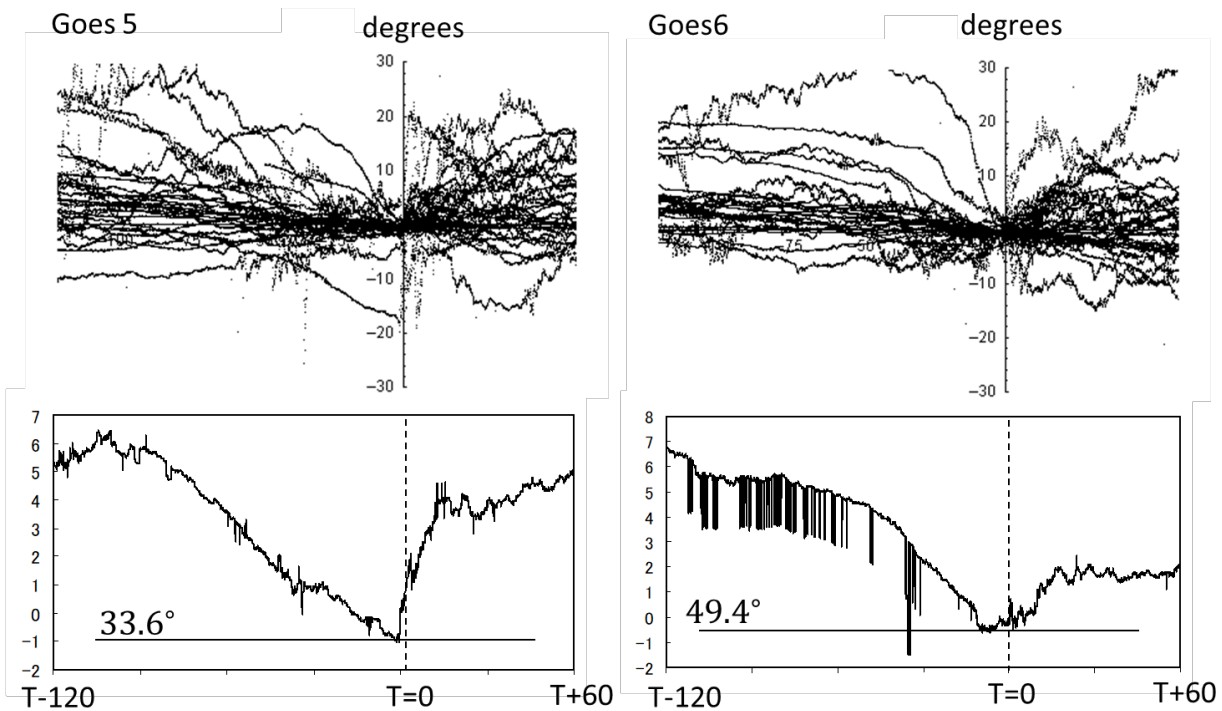

# Figure 1

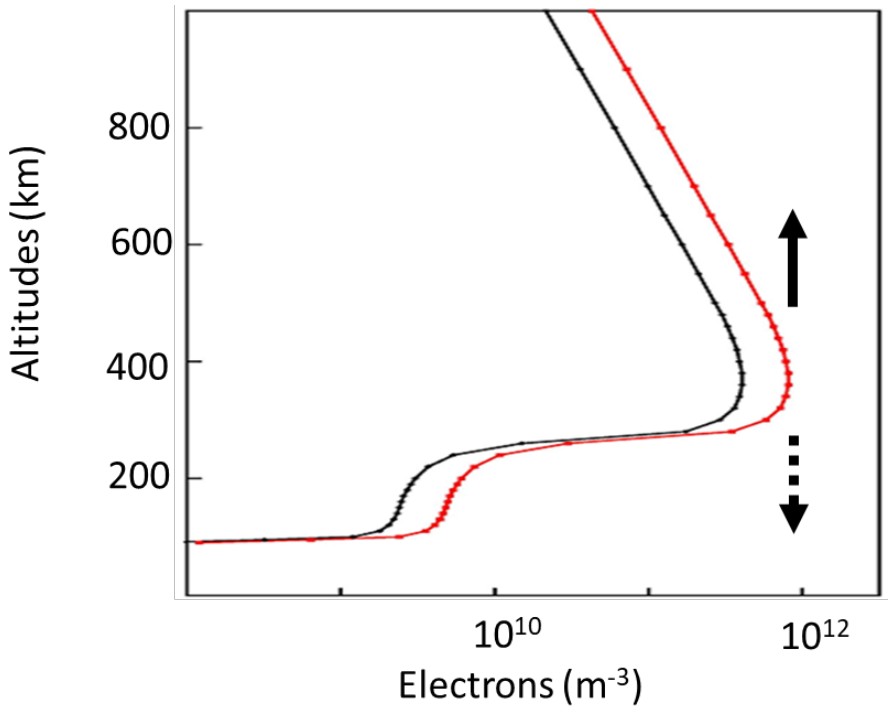

# Figure 2

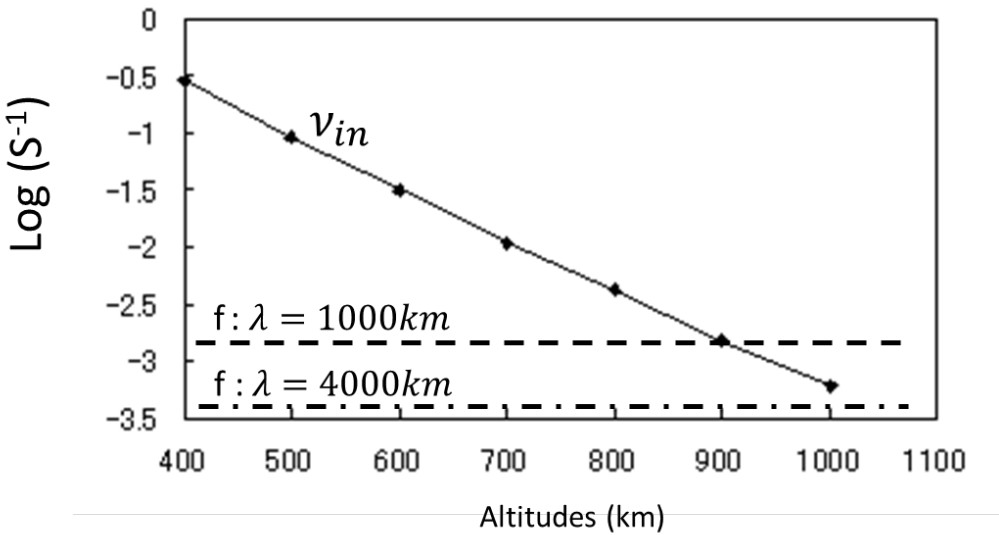

# Figure 3

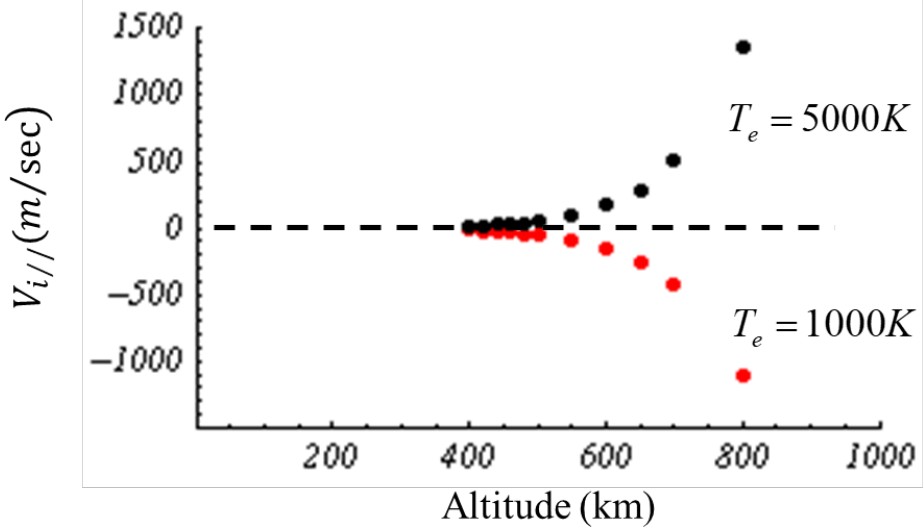

# Figure 4

(A)

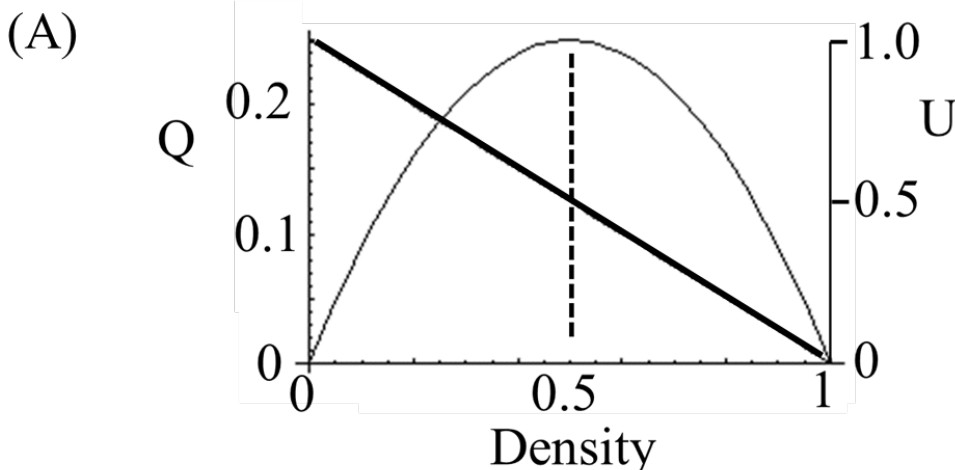

(B)

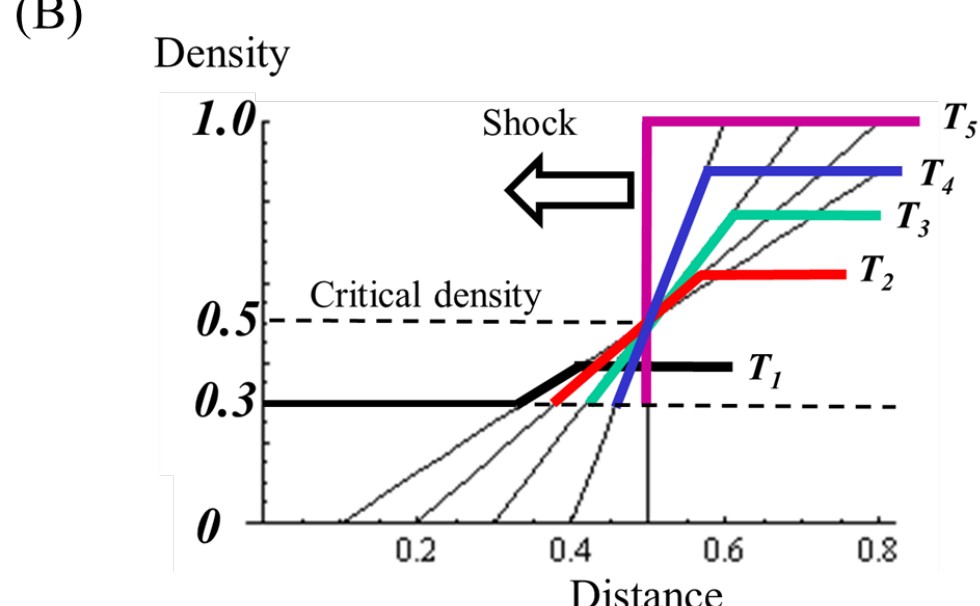

Figure 5