# Peer review of "A new scenario applying traffic flow analogy to poleward expansion of auroras"

_Annales Geophysicae, 2018_

## Referee Comment (RC1) · Anonymous Referee #1 · 12 Sep 2018

The paper by O. Saka attempts to address why aurora expands poleward but plasma flows equatorward during the substorm expansion phase. The author provides an analytical theory and concludes that poleward expansion is caused by the shock front of density accumulation propagating poleward. As commented below, the author's approach is oversimplified and omits critical nature of the ionosphere during the substorm expansion phase, such as magnetic field dipolarization, precipitation, and two dimensional system. Currently I don't think that the theory provided here gives a solid answer to the poleward expansion but need a more sophisticated approach.

The author did not consider that magnetic field geometry changes during the expansion phase. As magnetic fluxes accumulate on closed field lines due to reconnection during the substorm expansion phase (together with earthward flows known as BBF), the

magnetic field becomes more dipolar and the mapping location of a certain geocentric distance moves poleward. This geometrical change can easily explain the discrepancy between the poleward aurora motion and equatorward plasma drift. Please consider how mapping location changes by depolarizing magnetic field will contribute to author's story.

Equations (2) and (6) assume that there is no source term in the continuity equation. This assumption is not valid during the substorm expansion phase because of intense particle precipitation and vertical transport. Thus the density accumulation that the author obtained will be substantially modified. From this standpoint, the traffic flow analogy does not accurately represent the expansion phase. Please consider the effect of the source term.

The author also assumes the one dimensional system. Expansion phase aurora including surges is two dimensional, where the electric field converges to the center of surges [Opgenoorth et al., 1983]. The distance between equipotential lines becomes larger when the electric field decreases. In this situation the density does not pile up but spreads azimuthally when the electric field decreases. The one dimensional assumption does not consider this effect.

The author provided an equation for the shock front propagation but did not estimate if the speed is consistent with poleward expansion and if the critical density is within a realistic level of density in the ionosphere. Please make a quantitative assessment of this argument using realistic ionosphere parameters. Figure 4 only provides the parallel velocity but what's important for poleward expansion is the poleward velocity.
* * *

---

## Author Comment (AC1) · 1 Oct 2018

Our replies to Referee #1 are summarized as follows.

The poleward expansion of auroras was an auroral event observed in the initial pulse of Pi2 pulsations [Saka et al., JASTP, 80, 285-295, 2012]. The dipolarization of geomagnetic fields would have ended by the time of the Pi2 onset because Pi2 pulsations were triggered by the bifurcation of the fast earthward flows [Saka et al., JASTP, 72, 1100-1109, 2010]. The convection surge was produced by dawn-dusk stretching of flux tubes associated with longitudinal expansion of dipolarization [Saka and Hayashi, 2017].

We assumed that electron precipitations associated with the upward field-aligned currents lead to a temporal production of electron density in negative potential region. Quasi-neutrality is maintained because the Pedersen currents transported ions into this negative potential region. These effects enhance the ionospheric conductivities at the negative potential region. This may result in the ionospheric screening of the incident westward electric fields which was strengthened at the density peak. As a result, the ionosphere itself suppressed the further increase of the density at the peak by slowing down the directed flows. The flow velocity (U) is a function of ionospheric density (n) in such a way that U decreases with increasing n. This relation is consistent with the flux-density curve in traffic flow. Density accumulation may evolve nonlinearly following this curve.

In the E region, a large density gap occurred between ions and electrons by a mobility difference. These gaps were reduced by the Pedersen currents and retain quasi-neutrality. Figure 1 depicts an example of electrostatic potential pattern at T=4 (prior to the onset of poleward expansion) caused by ExB drift. Westward electric fields transmitted from the magnetosphere are assumed to extend in east-west directions by the form, $Ey=Exp[-(x/20)^2 - (y/40)^2]$. Here, x is in north-south and y is in east-west directions. Longitudinal scale is two times larger than that in latitudes. The ionospheric currents close to the field-aligned currents (upward from the negative potential region and downward towards the positive potential region) thereby sustaining these potential structures and retaining a quasi-neutral condition. The convergent electric fields in negative potential regions in the equatorward part expand poleward by the nonlinear evolution (T5 in Figure 2B). Though not shown in Figure 1, the transmitted westward electric fields skew potential contours in positive potential regions by moving the center towards the west. This skewing could be insignificant in the negative potential regions because electric fields from the magnetosphere were weakened.

A nonlinear evolution of the density accumulation with time is illustrated in Figure 2B using flux-density curve in Figure 2A. Density increased in a step-like manner, from T1 to T5. The shock formation started when the density increased over the critical density.

The velocity of the shock is the same as the flow velocity towards downstream, namely 2.0 km/s for 100mV/m of the westward electric field amplitudes, which is a typical value of the expansion velocity associated with Pi2 pulsations [Saka et al., JASTP, 80, 285-295, 2012]. The critical density separates free flows (lower density than the critical density) and congested flows (higher density than the critical density). It is supposed that the free flows dominated in the westward electric field region when they propagated equatorward, while congested flows dominated when they stopped or even reversed its propagation direction. The shock formation delayed in the former case while it starts quickly for the latter case.

Major losses along the field lines may be caused by steady-state plasma motions existing up to 800km in altitudes, where ion velocity exceeded the phase velocity of ion acoustic wave. Field-aligned velocities of the steady-state ion flows varied from 15m/s at 400km to 1367 m/s at 800km in altitudes. The upward ion flux carried by the flow also varied from $5.9 \times 10^{12} m^{-2} s^{-1}$ at 400km to $8.2 \times 10^{13} m^{-2} s^{-1}$ at 800km in altitudes. This means that the steady-state flows evacuated the accumulated density in 250 s (4.2 min) at 750 km, while ions at 450 km were slowly evacuated to the higher altitudes in 3300 s (55 min). If we consider a transient event of few minutes, field-aligned loss may become negligible below 500 km. In ten seconds, losses may be negligible throughout the topside ionosphere.

[Figure]

**Fig. 1.** Contour plots of electrostatic potential at T=4 (before onset) in the E region produced by the two-dimensional westward electric fields.

[Figure]

**Fig. 2.** (A) flux (q)-density (n) curve represented by q=n(1-n). (B) Plots of nonlinear evolution of density accumulation.

---

## Referee Comment (RC2) · Anonymous Referee #2 · 10 Oct 2018

In the current manuscript, a new scenario is proposed to explain how the ionospheric drift directions can be equatorward within the activated expansion-phase auroras while the poleward regions expand poleward. The main idea proposed is that if one takes into account compressibility effects in the ionosphere, a sequence of events can potentially occur in which density accumulations in the ionosphere end up producing field aligned currents that propagate poleward. While the concept is interesting and is described in some detail, the manuscript does not present a self-consistent treatment in terms of real MI-coupling processes (no model really does this properly yet) and does not simulate any events that can be validated against observations. The simulations are fairly simplistic with questionable assumptions. For example, the evolution of density in equation (2) is treated as a 1D problem based on the assumption that the imposed

convection surge spreads more widely in longitude than in latitude. In addition, there is no model for how the "convection surge" is created or what might be going on in the tail that led to its creation. In real substorms, the magnetic field is highly variable including a slower stretching of the field during the growth phase followed by a more rapid and typically complex dipolarization phase. These B-field variations will change the mapping in a time-dependent manner and produce a complex transient reponse in the form of MI-coupling which is not accounted for here. It is therefore difficult to gauge how successful the model really is in terms of describing real substorms.

A major motivation of the manuscript appears to be the assumption that the direction of plasma drifts in the ionosphere during substorms is not understood. The author points out that the poleward expansion of the auroras [associated with substorms] is opposite to the general motion of plasma drift (or auroral forms?) within the expanding [bulge] (e.g. Lines 30-36 of the manuscript). As the author notes, this has been known for a very long time. Although it was not really discussed in the original phenomenological model of Akasofu [1964], it was certainly known early on. See, for example, Figure 3 of Akasofu et al. [1966]. That said, the author probably should not claim that this "discrepancy" has been a source of ongoing debate since that time.

Global auroral observations and ground-based all sky imagery clearly shows that while the envelope of activity expands poleward during the expansion phase, forms within the bulge tend to move equatorward. The Earthward motion (often in the form of streamers) is completely consistent with the virtually universally-accepted idea that convection in the nightside magnetotail is Earthward (on the sunward side of reconnection sites) and that it is typically bursty in nature (i.e. bursty bulk flows described by Baumjohan, Angelopolous, etc..). The poleward motion of the expanding envelope is also completely consistent with the progression of the substorm x-line toward the lobe field lines in the tail (and associated plasmoid release).

The controversy addressed by THEMIS was not really about which direction auroral forms (or ionospheric plasma drifts) move during the expansion phase, but rather which

region activates first (i.e. inside-out or outside-in) around a much narrower time period near the onset of a substorm. The controversies revolve around: (1) why in the outside-in model, auroral signatures of a higher latitude reconnection process are not clearly observed prior to the lower-latitude activation and subsequent poleward-expanding envelope, and (2) how in the inside-out model, a near-Earth disturbance is related to activation of mid-tail reconnection that is known to develop during substorms (and whose signatures have sometimes been reported to precede the lower latitude activation.)

Some major issues with the manuscript are:

(1) It appears to ignore mechanisms in the magnetotail that are already fairly well understood. For example, it is known that a new x-line is formed in the magnetotail sometime early on during substorms. In addition, it is known that convection from both the pre-existing x-line and the new substorm x-line proceeds in a bursty manner which produces Earthward-directed bursty bulk flows (BBFs). There has also been considerable work done in showing how such localized flows relate to equatorward-moving auroral forms in the ionosphere (i.e. streamers) and particle energization in the magnetosphere.

(2) Intensifications of the poleward expanding edge of the bulge are often accompanied by equatorward ejection of streamer forms. How does the current model account for this type of activity? This is very easy to explain in the context of reconnection, but it is not clear how the current scenario would acount for such observations.

(3) Low-altitude observations have shown that the auroras at the poleward edge of the expanding disturbance are Alfvenic in nature (i.e. due to wave-accelerated electrons) (e.g. see Mende, (2016)). This is in contrast to more equatorward auroral arcs that are often associated with inverted-V type potential structures (acceleration of electrons through a field-aligned potential drop). It is not clear how the current model fits these observations. How does the model produce broadband wave acceleration of auroral particles at the poleward edge and inverted-V type structure in the more equatorward

regions? The current scenario discusses ion acoustic waves travelling upward and downward along field lines in the ionosphere. Are these observed? How do they relate to observations of Alvfenic auroras already published?

(4) The poleward-expanding bulge during a substorm eventually grows to coalesce with the open-closed boundary, while at the same time plasmoids are known to be released in the more distant tail. This is exceptionally strong evidence that new reconnection at a mid-tail site is a dominant driver during the expansion phase. It is not clear how the current model is consistent with these observations.

(5) The model assumes the imposition of physical processes and dynamics from the magnetotail in the form of a substorm "convection surge" at the dipolarization onset, but then argues that the ionosphere ends up directly driving processes in the tail (e.g. lines 199-202). It is probably more appropriate to think of the entire process via a more unified MI-coupling approach, but the fact that magnetotail processes appear to be required to drive the effects considered in the current model, indicates that the magnetosphere is really the source of driving in the model.

(6) The manuscript should have described other models for poleward expansion of the aurora (as described in some of the points above), but one in particular has a very similar analogy to propagation of brake lights on cars in traffic. Specifically, the flow-braking model postulates a very similar tailward-propagating distrubance that could be related to poleward propagation in the ionosphere. Especially since the language used for these models is quite similar, the author should really have provided some discussion of it here.

In summary, the proposed model may be valuable in understanding some of the details of MI-coupling that occurs near onset, but it is very unlikely to be able to explain the full wealth of observations that are known to exist in the ionosphere and in the tail during expansion phase. In this view, the proposed scenario should not really be considered as an alternative to other models, but rather as a refinement on them. Otherwise,

the author should be able to explain why all of the other mechanisms proposed in the past somehow cease to operate as a result of the currently proposed concept. For example, we know that near-Earth reconnection occurs during substorms. How does the current model stop this reconnection site from imposing a poleward-propagating envelope of activity in the ionosphere as it progresses toward the lobe field lines in the tail? Similarly, the manuscript does not address how the model is consistent with the vast array of known observations of auroras during substorms (e.g. how does it produce Alfvenic auroras at the poleward edge of the expanding bulge? How are streamers produced? etc.)

I would reccomend that the focus of the paper should not be an all-encompassing alternative model for poleward expansion during the expansion phase in general (this is how the paper currently reads), but rather that it merely point out that some MI-coupling phenomena associated with ionospheric compressibility may be important during the initial onset phase of substorms.

References:

Akasofu, S. (1964), The development of the auroral substorm, Planet. Space. Sci., 12, 273.

Akasofu, S., D. S. Kimball, and C. Meng (1966), Dynamics of the aurora – vii equatorward motions and the multiplicity of auroral arcs, Journal of Atmospheric and Terrestrial Physics, 28, 627–635.

Mende, S. B. (2016), Observing the magnetosphere through global auroral imaging: 1. Observables, J. Geophys. Res. Space Physics , 121 , 10623–10637, doi:10.1002/2016JA022558

---

## Author Comment (AC2) · 19 Oct 2018

Our replies to Referee #2 are summarized as follows.

The first 10 min interval of Pi2 onset is a transitional state of the substorm dominated by MHD processes associated with earthward flow and its bifurcation [Saka et al., JASTP, 72,1100-1109, 2010]. In this interval, a collapse of dipolarized geomagnetic field configurations started to return to the original field line configurations characterized by the potential fields. The poleward expansion of auroras occurred in the first one-minute-interval of Pi2 onset (initial pulse of Pi2). [Saka et al., JASTP, 80, 285-295, 2012]. During the subsequent Pi2 pulses following the initial pulse, an auroral surge propagating eastward or westward at the poleward boundary of the auroral zone was

observed in all-sky images that were interpreted as an auroral manifestation of flow bifurcation of fast earthward flows (BBFs) [Saka et al., JASTP, 80, 285-295, 2012].

It is supposed that auroral signatures associated with the BBFs could be found in low latitudes because the inclination angle of field lines in the equatorial plane attained minimum values at the Pi2 onset (see Fig1 attached to this reply, reproduced from [Saka et al., JASTP, 2010]), that were caused by the continuous inflows towards the equatorial plane in the preonset intervals [Saka and Hayashi, JASTP, 164, 235-242, 2017]. This means that an activation of the preonset auroras can be regarded as an auroral manifestation of BBFs. The activation was followed by auroral beading and dawn-dusk stretching of the flux tubes. In contrast, an evolution of N-S auroras (streamers) from the poleward auroral boundary occurred in association with Pc4 pulsations and, at least at the geosynchronous orbit, no inflows were observed [Saka et al., JASTP, 145, 114-124, 2016].

At the poleward boundary of auroral zone, fast aurora surge correlating to Pi2 pulse as described above was observed in 66 – 74 ILat. Velocity of the surge, 12 - 30km/s at 100 km in altitudes, corresponds to azimuth flows of the order of 190 – 480 km/s at L=8.5 (70 ILat). It is supposed that these azimuthal flows may be an energy source exciting Alfven waves in the poleward boundary of auroral zone. Steady state parallel ion flows created by the ion acoustic waves along the compressed collisional ionosphere carry ionospheric ions towards the topside ionosphere as depicted in Figure 4B. The upward ion flux carried by the flow varied from $5.9 \times 10^{12} m^{-2} s^{-1}$ at an altitude of 400km to $8.2 \times 10^{13} m^{-2} s^{-1}$ at 800km. This means that the parallel flows evacuated the accumulated ions in 250 s (4.2 min) at 750 km, while ions at 450 km were slowly evacuated to the higher altitudes in 3300 s (55 min). Quick evacuation of ions in higher altitudes may create ion holes in the topside ionosphere. These ion holes in the topside ionosphere are potential candidates for double layers, an inverted-V potential structure [Hudson et al., JGR, 88, 916-926, 1983]. Ion acoustic waves related to auroral arcs were observed in the EISCAT radar echoes as naturally enhanced ion acoustic spectral shoulders [Wahlund et al., JGR, 97, 3019-3037, 1992]. In the present scenario, poleward boundary aurora surge is Alfvenic in nature and auroras associated with the ionospheric compression are of the inverted-V type.

In nightside magnetosphere, Pi2 pulsations are regarded as a repeating flow bifurcation with a time constant of minute, or flow diversion of BBF substructures [Saka et al., JASTP, 72,1100-1109, 2010]. If BBF substructures are related somewhat to the substorm cadence, Pi2 pulsations are geomagnetic manifestation of mid-tail reconnection repeated with the time constant of one minute.

In the initial pulse of Pi2, flux tubes stretching in tailward directions switched to dawn-dusk directions through MHD processes. The dawn-dusk stretching in the initial pulse was not localized but expanded in longitudes over 19 to 04 LT [Saka et al., JASTP, 62, 17-30, 2000]. A possible candidate for the MHD processes is an excitation of slow MHD waves [Saka and Hayashi, JASTP, 164, 235-242, 2017]. This switching from tailward to dawn-dusk directions disrupts the cross-tail currents and may produce convection surge and associated westward electric fields in the midnight sector [Saka et al., Ann Geophys., 32, 1011-1023, 2014]. The convection surge may not repeat in the same longitudes. This means that auroral expansion was observed once in the initial pulse of Pi2 pulsation.

We proposed the traffic flow analogy to explain the poleward expansion of auroras because the flux-density curve (Figure 3A) for the traffic flow was applicable to the ionospheric case. A similar case may be found in "flow braking in near-Earth tail" [Shiokawa et al., GRL, 24, 1179-1182, 1997] if an excess pileup of the pressures was suppressed by the ion flow itself as expressed in the advective term in the equation of motion. This may minimize the velocity of the ion flows at the pressure peak analogous to a tailback of cars in traffic flows.
* * *
[Figure]

**Fig. 1.** Mean inclination angle of geomagnetic fields at geosynchronous orbit (Goes5) from T0 -120 min to T0 +60 min, where T0 denotes Pi2 onset.

---

## Author Response (AR1)

To Dr. Gkioulidou, Topical Editor

The submitted manuscript has been substantially revised by adding new sections 2 and 5 thereby increasing the number of pages from 13 to 18. Each issue has been addressed in the prescribed Comment/Response format. These responses were marked in the manuscript in bold and were specified by line numbers.

Point-by-point replies to the comments raised by Referee #1, a list of all relevant changes and marked-up manuscript are given below.

**Comment 1:**
"The author did not consider that magnetic field geometry changes during the expansion phase. As magnetic fluxes accumulate on closed field lines due to reconnection during the substorm expansion phase (together with earthward flows known as BBF), the magnetic field becomes more dipolar and the mapping location of a certain geocentric distance moves poleward. This geometrical change can easily explain the discrepancy between the poleward aurora motion and equatorward plasma drift. Please consider how mapping location changes by dipolarizing magnetic field will contribute to author's story."

Response:
Poleward expansion of auroras arising out of the onset arc was observed in the initial pulse of Pi2 pulsations [Saka et al., 2012]. Statistical study of geomagnetic fields at geosynchronous orbit during Pi2 showed that field line inclination at geosynchronous orbit (Goes5/6 at $285^\circ$/$252^\circ$ in geographic coordinates) decreased continuously in the growth phase and attained $33.6^\circ$/$49.4^\circ$ in dipole coordinates 2-min prior to the initial peak of Pi2 amplitudes (see Figure A below, reproduced from Saka et al., 2010). This result suggests the observed field line inclination prior to the onset conforms to T89 model of Kp=4 [Tsyganenko, 1989]. These field lines were mapped to auroral ionosphere at $63.4^\circ$N/$62.7^\circ$N in geomagnetic coordinates corresponding to a latitude of onset arc. From these estimations, we suggest that Bursty Bulk Flows (BBFs) appeared first at the onset latitudes and activated preonset auroras. In the initial pulse of Pi2 pulsations, field line inclination of Goes5 increased in a step-like manner, while for Goes6 transient pulses were observed [Saka et al., 2010]. The average latitudes of Goes5 and Goes6 were at $10.3^\circ$N and $7.9^\circ$N, respectively in the T89 model for the average Kp index of 3. It is likely that dipolarization was composed of transient pulses at latitudes closer to the equatorial plane. In the following Pi2 pulses, an auroral surge was observed in all-sky images between 66 $^{\circ}$ N to 74 $^{\circ}$ N in geomagnetic latitudes. They propagated eastward or westward at the poleward boundary of the auroral zone and were interpreted as an auroral manifestation of flow bifurcation of fast earthward flows (BBFs) [Saka et al., 2012].

[Figure]

Figure A: Inclination angle in degrees measured positive northward from the V-D plane from 120 min prior to the Pi2 onset (T-120) and to 60 min after the Pi2 onset (T+60) reproduced from Saka et al. (2010). Magnetometer data of Goes 5/6 were represented in HVD coordinates, H is positive northward parallel to dipole axis, V is radial outward, and D is dipole east. Epoch superposition of 30 Pi2 events and mean angles calculated from them are plotted in top and in lower panels, respectively. Mean inclination angle at 2-min before the initial peak of Pi2 amplitudes was 33.6 $^{\circ}$ for G5 and 49.4 $^{\circ}$ for G6 in dipole coordinate. Dipolarization (T=0) was step-like at 10.3 $^{\circ}$ N, while at 7.9 $^{\circ}$ N it was composed of transient pulses.

Simultaneous with reconfigurations of field lines in meridional plane, field lines stretching initially in tailward directions switched to dawn-dusk directions in the initial pulse of Pi2s [Saka et al., 2000]. This switching was associated with the excitation of slow MHD wave by the increasing inflows of plasma sheet ions toward the equatorial plane in growth phase [Saka and Hayashi, 2017]. An alternative explanation for polarization switching may involve Ballooning instability of the coupled poloidal Alfven and slow magnetoacoustic modes [Rubtsov et al., 2018]. This switching from tailward to dawn-dusk directions disrupts the cross-tail currents creating a step like dipolarization or dipolarization pulses in the initial pulse of Pi2 pulsations and may produce convection surge and associated westward electric fields in the midnight sector.

Please refer to Lines 43 – 80 for more detail.

**Comment 2:**

"Equations (2) and (6) assume that there is no source term in the continuity equation. This assumption is not valid during the substorm expansion phase because of intense particle precipitation and vertical transport. Thus the density accumulation that the author obtained will be substantially modified. From this standpoint, the traffic flow analogy does not accurately represent the expansion phase. Please consider the effect of the source term."

Response:

For the equatorward drift in the flow channel to be an order of kilometers per second corresponding to E=100 mV/m in auroral ionosphere, both outflows and precipitation may not bring significant changes to the flux carried by $E \times B$ drift in the flow channel. We then approximate one dimensional (along the drift path) conservation equation in the flow channel.
Please refer to Lines 124 – 174, and 177 – 189 for more detail.

**Comment 3:**

"The author also assumes the one dimensional system. Expansion phase aurora including surges is two dimensional, where the electric field converges to the center of surges [Opgenoorth et al., 1983]. The distance between equipotential lines becomes larger when the electric field decreases. In this situation the density does not pile up but spreads azimuthally when the electric field decreases. The one dimensional assumption does not consider this effect."

Response:

We assume that the westward electric fields generated in the magnetosphere in the initial pulse of Pi2 pulsations were transmitted along the field lines to the auroral ionosphere by the guided poloidal mode [Radoski, 1967] from the surge location and created an equatorward flow in the auroral ionosphere. The flows are assumed to be confined in the flow channel expanding in the north-south directions in the midnight sector. The low-latitude end of the flow channel was at the latitudes of the onset arc corresponding to earthward edge of the BBFs. The high-latitude end may not expand beyond the poleward boundary of auroral zone. Longitudinal width of the flow channel may be in about 1 to 2 hours in local time (~1000 km along $65^{\circ}$N) corresponding to horizontal scale size of Pi2 vortices [Saka et al., 2014]. In this flow channel, two-dimensional potential structure grows.

In order to interrupt the excess accumulation in the flow channel, we suggested that ionosphere would have responded nonlinearly to decelerate the equatorward drifts in the flow channel. These were introduced by the ionospheric screening of the penetrated (total) electric fields. In this process, accumulated plasmas would have expanded backward while they were trapped in the flow channel.

Please refer to Lines 83 – 92 for more detail.

**Comment 4:**

"The author provided an equation for the shock front propagation but did not estimate if the speed is consistent with poleward expansion and if the critical density is within a realistic level of density in the ionosphere. Please make a quantitative assessment of this argument using realistic ionosphere parameters."

Response:

From the ionospheric screening process, we tentatively assume that flow velocity $U$ is a function of the density $n$. Then the conservation equation along the flow channel may be written as,

$$\frac{\partial n}{\partial t} + \frac{\partial}{\partial x} Q(n) = 0 . \qquad (9)$$

Here, $Q(n)$ is a mass flux defined by $Q(n)=nU(n)$. This relation can be reduced to nonlinear wave equation,

$$\frac{\partial n}{\partial t} + c(n)\frac{\partial n}{\partial x} = 0 . \qquad (10)$$

Here $c(n)$ is a wave propagation velocity defined by $c(n) = U(n) + nU'(n)$, $U(n)$ is a drift velocity in the flow channel, and $U'(n)$ denote braking/acceleration of the drift velocity by increasing and decreasing density. The equation (10) is often referred to as propagation of "kinematic waves" to describe traffic flow [Lighthill and Whitham, 1955]. In the following, we use dimensionless units normalized by $U_m$, and $n_m$. Here, $U_m$ and $n_m$ denote maximum drift velocity at $n=0$, and maximum density for the complete stops of the drift, respectively. Assuming a constant braking in the flow channel, we define $U$ by a linear function of density $n$ as $U(n)=1-n$. Noting that $Q'(n)=c(n)$, this relation is reduced to the equation, $Q(n)=n(1-n)$, identical to the case for the traffic flow [Whitham, 1999]. Both the $U$ and $Q$ are plotted in Figure 4A as a function of $n$. A nonlinear evolution of the waves is presented in Figure 4B by the characteristic curves. In the case of the traffic, the initial flows started from $n=0$ and stopped at $n=1.0$ by the tailback of cars. For the case of the ionosphere, the ionospheric density started from a finite density, $n=0.3$ (Figure 4B), and increased to $n=1.0$ to terminate the flow by the full screening. The nonlinear evolution of the density profile in time is shown in Figure 4B in colors from black ($T=T_1$), red ($T=T_2$), green ($T=T_3$), blue ($T=T_4$), and to purple ($T=T_5$). After $T=T_5$, the waves propagate upstream (poleward) as a shock. The shock velocity, $V$, is given as [Whitham, 1999],

$$V = \frac{Q(n_2) - Q(n_1)}{n_2 - n_1} . \qquad (11)$$

Here, subscript 1 is for the values ahead shock and subscript 2 is for the values behind. Noting that

$Q(n_2)=0$ and substituting $Q(n_1)=n_1(n_2-n_1)$, the equation (11) can be reduced to $V=-n_1$ in dimensionless unit. The propagation velocity of the shock is related to the densities ahead. For the case of $n=0.3$ in Figure 4B, shock velocity can be estimated to be $-0.3U_m$. Here, $U_m$ denotes maximum drift velocity in the ionosphere where ionospheric screening effects vanished by the condition, $\Sigma_P/\Sigma_A \ll 1$.

Please refer to Lines 190 – 238 for more detail.

**Comment 5:**

"Figure 4 only provides the parallel velocity but what's important for poleward expansion is the poleward velocity."

Response:

We estimated parallel flow velocity generated by the transient accumulation in the flow channel. It may excite ion acoustic wave traveling upward from the peak of the F layer. A steady-state ion flows associated with the ion acoustic wave exist in the topside ionosphere and contribute to the vertical transport of materials from the ionosphere. This result was compared with the horizontal flows in the flow channel.

Please refer to Lines 124 -174.

Point-by-point replies to the comments raised by Referee #2, a list of all relevant changes and marked-up manuscript are given below.

**Comment 1:**

"In the current manuscript, a new scenario is proposed to explain how the ionospheric drift directions can be equatorward within the activated expansion-phase auroras while the poleward regions expand poleward. The main idea propose is that if one takes into account compressibility effect in the ionosphere, a sequence of events can potentially occur in which density accumulations in the ionosphere end up producing field aligned currents that propagate poleward. While the concept is interesting and is described in some detail, the manuscript does not present a self-consistent treatment in terms of real MI-coupling processes (no model really does this properly yet) and does not simulate any events that can be validated against observations. The simulations are fairly simplistic with questionable assumptions. For example, the evolution of density in equation (2) is treated as a 1D problem based on the assumption that the imposed convection surge spreads more widely in longitude than latitude. In addition, there in no model for how the "convection surge" is created or what might be going on in the tail that led to its creation. In real substorms, the magnetic field is highly variable including a slower stretching of the field during the growth phase followed by a more rapid and typically complex dipolarization phase. These B-field variations will change the mapping in a time dependent manner and produce a complex transient response in the form of MI-coupling which is not accounted for here. It is difficult to gauge how successful the model really is in terms of describing real substorms."

Response:
We added a new section with the title "Auroras and field line reconfigurations associated with Pi2" in Lines 43 - 80. This new section may respond to some extent to the comment above.

**Comment 2:**

"A major motivation of the manuscript appears to be the assumption that the direction of plasma drifts in the ionosphere during substorms is not understood. The author points out that the poleward expansion of the auroras [associated with substorms] is opposite to the general motion of plasma drift (or auroral forms?) within the expanding [bulge] (e.g. Lines 30-36 of the manuscript). As the author notes, this has been known for a very long time. Although it was not really discussed in the original phenomenological model of Akasofu et al. [1966]. That said, the author probably should not claim that this "discrepancy" has been source of ongoing debates since that time".

Response:

Sentences in the first version in Lines 30 – 36 were changed to:

"Plasma drifts in the ionosphere observed by the balloon-measured electric fields [Kelley et al., 1971], by the Ba releases [Haerendel, 1972] and by radar observations [Nielsen and Greenwald, 1978] did not match the expanding trajectories of auroras. This fact raises one of several unanswered questions involving violent poleward motion of auroras [Akasofu et al., 1966]. To account for the difference in propagation directions, it was suggested that auroras were directly connected to the reconnecting flux tube moving tailward in the plasma sheet. This idea suggests that the primary sources of particles are in the magnetosphere, though they might be accelerated in lower altitudes for precipitations. This concept has been accepted in the literature for many decades."

Please refer to Lines 23 -31.

**Comment 3:**

"Global aurora observations and ground-based all sky imagery clearly shows that while the envelop of activity expands poleward during the expansion phase, forms within the bulge tend to move equatorward. The Equatorward motion (often in the form of streamers) is completely consistent with the virtually universally-accepted idea that convection in the nightside magnetotail is Earthward (on the sunward side of reconnection sites) an that it is typically bursty in nature (i.e. bursty bulk flows described by Baumjohan, Angelopoulos, etc..). The poleward motion of the expanding envelop is also completely consistent with the progression of the substorm x-line toward the lobe field lines in the tail (and associated plasmoid release)."

Response:

Please refer our replies to Comment 6 for a response to the above comment.

**Comment 4:**

"The controversy addressed by THEMIS was not really about which direction auroral forms (or ionospheric plasma drifts) move during the expansion phase, but rather which region activates first (i.e. inside-out or outside-in) around a much narrower time period near the onset of a substorm. The controversies revolve around: (1) why in the outside-in model, auroral signatures of a higher latitude reconnection process are not clearly observed prior to the lower-latitude activation and subsequent poleward-expanding envelop, and (2) how in the inside-out model, a near-Earth disturbance is related to activation of mid-tail reconnection that is known to develop during substorms (and whose signatures have sometimes been reported to precede the lower latitude activation.)"

Response:

Please refer our replies to Comment 5 for a response to the above comment.

**Comment 5:**

"It appears to ignore mechanisms in the magnetotail that are already fairly well understood. For example, it is known that a new x-line is formed in the magnetotail sometime early on during substorms. In addition, it is known that convection from both the pre-existing x-line and the new substorm x-line proceeds in a bursty manner which produces Earthward-directed bursty bulk flows (BBFs). There has also been considerable work done in showing how such localized flows relate to equatorward-moving auroral forms in the ionosphere (i.e. streamers) and particle energization in the magnetosphere."

Response:

Poleward expansion of auroras arising out of the onset arc was observed in the initial pulse of Pi2 pulsations [Saka et al., 2012]. Statistical study of geomagnetic fields at geosynchronous orbit during Pi2 showed that field line inclination at geosynchronous orbit (Goes5/6 at $285^\circ$/$252^\circ$ in geographic coordinates) decreased continuously in the growth phase and attained $33.6^\circ$/$49.4^\circ$ in dipole coordinates 2-min prior to the initial peak of Pi2 amplitudes (see Figure A below, reproduced from Saka et al., 2010). This result suggests the observed field line inclination prior to the onset conforms to T89 model of Kp=4 [Tsyganenko, 1989]. These field lines were mapped to auroral ionosphere at $63.4^\circ$N/$62.7^\circ$N in geomagnetic coordinates corresponding to a latitude of onset arc. From these estimations, we suggest that Bursty Bulk Flows (BBFs) appeared first at the onset latitudes and activated preonset auroras. In the initial pulse of Pi2 pulsations, field line inclination of Goes5 increased in a step-like manner, while for Goes6 transient pulses were observed [Saka et al., 2010]. The average latitudes of Goes5 and Goes6 were at $10.3^\circ$N and $7.9^\circ$N, respectively in the T89 model for the average Kp index of 3. It is likely that dipolarization was composed of transient pulses at latitudes closer to the equatorial plane. In the following Pi2 pulses, an auroral surge was observed in all-sky images in $66^\circ$N to $74^\circ$N in geomagnetic latitudes. They propagated eastward or westward at the poleward boundary of the auroral zone and were interpreted as an auroral manifestation of flow bifurcation of fast earthward flows (BBFs) [Saka et al., 2012].

Please refer to Lines 43 – 80 in detail for new section titled "Auroras and field line reconfigurations associated with Pi2".

[Figure]

Figure A: Inclination angle in degrees measured positive from the V-D plane from 120 min prior to the Pi2 onset (T-120) and to 60 min after the Pi2 onset (T+60) reproduced from Saka et al. (2010). Magnetometer data of Goes 5/6 were represented in HVD coordinates, H is positive northward parallel to dipole axis, V is radial outward, and D is dipole east. Epoch superposition of 30 Pi2 events and mean angles calculated from them are plotted in top and in lower panels, respectively. Mean inclination angle at 2-min before the initial peak of Pi2 amplitudes was 33.6° for G5 and 49.4° for G6 in dipole coordinate. Dipolarization (T=0) was step-like at 10.3°N, while at 7.9°N it was composed of transient pulses.

**Comment 6:**

"Intensifications of the poleward expanding edge of the bulge are often accompanied by equatorward ejection of streamer form. How does the current model account for this type of activity? This is very easy to explain in the context of reconnection, but it is not clear how the current scenario would account for such observations."

Response:

Pi2 auroras were triggered by the onset of slow MHD wave excited by the inflows (Poynting flux) of plasma sheet ions towards the equatorial plane. In contrast, an evolution of N-S auroras (streamers) from the poleward auroral boundary occurred in association with a local enhancement of cross-tail electric fields but no inflows were observed that leads to the excitation of slow MHD wave [Saka et al., 2016]. These streamers may be observed in the intervals prior to the Pi2 onset [Nishimura et al., 2011]. In our scenario, streamers may not trigger the Pi2 onset.

Please refer to Lines 61 – 80 in detail.

**Comment 9:**

"The model assumes the imposition of physical processes and dynamics from the magnetotail in the form of a substorm "convection surge" at the dipolarization onset, but then argues that the ionosphere ends up directly driving processes in the tail (e.g. lines 199-202). It is probably more appropriate to think of the entire process via a more unified MI-coupling approach, but the fact that magnetotail processes appear to be required to drive the effects considered in the current model, indicates that the magnetosphere is really the source of driving in the model."

Response:

In the initial pulse of Pi2, flux tubes stretching in tailward directions switched to dawn-dusk directions through MHD processes. A possible candidate for the MHD processes is an excitation of slow MHD waves. This switching from tailward to dawn-dusk directions disrupts the cross-tail currents and produced convection surge and associated westward electric fields in the midnight sector. This MHD process (or instability) triggered the onset in the magnetosphere. In response to the onset, the ionosphere created parallel potential in the topside ionosphere which drove ion outflows, electron precipitations and upward field-aligned currents. A part of the upward field-aligned currents originated from the ionosphere may be observed at the geosynchronous orbit in the first 10 min intervals of Pi2 onset (see Figure B below).

[Figure]

Figure B: Azimuth angle in degrees measured in the V-D plane positive counterclockwise from the V-axis from 120 min prior to the Pi2 onset (T-120) and to 60 min after the Pi2 onset (T+60) reproduced from Saka et al. (2010). Magnetometer data of Goes 5/6 were represented in HVD coordinates, H is positive northward parallel to dipole axis, V is radial outward, and D is dipole east. The azimuth angle is $180°$ when the field line vectors point earthward. Epoch superposition of 30 Pi2 events and mean angles calculated from them are plotted in top and in lower panels, respectively. Mean azimuth angle at 2-min before the initial peak of Pi2 amplitudes (T=0) was $183°$ for G5 and $180°$ for G6 in dipole coordinate. In the first 10 min intervals of Pi2 onset marked by red arrows, field line shears were observed between G6 and G5 latitudes. The shear is consistent with the field line change associated with the upward field-currents between the latitudes of G5 and G6 at $10.3°$N and $7.9°$N, respectively, in the T89 model for the average Kp index of 3.

Please refer to Lines 61 – 80 for more detail.

**Comment 10:**

"The manuscript should have described other models for poleward expansion of the aurora (as described in some of the points above), but one in particular has a very similar analogy to propagation of brake lights on cars in traffic. Specifically, the flow-braking model postulates a very similar tailward-propagating disturbance that could be related to poleward propagation in the ionosphere. Especially since the language used for these models is quite similar, the author should really have provided some discussion of it here".

Response:

We proposed the traffic flow analogy to explain the poleward expansion of auroras because of the similarities in nonlinear processes in the flows. A similar case may be found in "flow braking in near-Earth tail" [Shiokawa et al., 1997].

Please refer to Lines 239 – 243.

**Comment 11:**

"In summary, the proposed model may be valuable in understanding some of the details of MI-coupling that occurs near onset, but it is very unlikely to be able to explain the full wealth of observations that are known to exist in the ionosphere and in the tail during expansion phase. In this view, the proposed scenario should not really be considered as an alternative to other models, but rather as a refinement on them. Otherwise, the author should be able to explain why all of the other mechanisms proposed in the past somehow cease to operate as a result of the currently proposed concept. For example, we know that near-Earth reconnection occurs during substorms. How does the current model stop this reconnection site from imposing a poleward-propagating envelop of activity in the ionosphere as it progresses toward the lobe field lines in the tail? Similarly, the manuscript does not address how the model is consistent with the vast array of known observations of aurora during substorms (e.g. how does it produce Alfvenic auroras at the poleward edge of the expanding bulge? How are streamer produced?)"

Response:

For our cases a new approach to auroral substorms developed from organizing the substorm processes in terms of the Pi2 pulsations observed in dip-equator; Huancayo in Peru and Christmas Island in Central Pacific. Low-latitude observations of geomagnetic Pi2 pulsations are believed to have some advantages in sorting out substorm cadence from the temporarily and spatially complexed processes in the magnetosphere.

[revised manuscript text omitted]

Figure 1

[Figure]

Figure 2

[Figure]

Figure 3

(A)

[Figure]

(B)

[Figure]

Figure 4

---

## Author Response (AR2)

To Dr. Gkioulidou, Topical Editor

I very much appreciate the critical review of our manuscript. Substantial revisions have accordingly been made. The following Comment/Response format addresses each issue. Relevant changes in the manuscript are denoted in bold type with line numbers.

Point-by-point replies to the comments raised by Referee, a list of all relevant changes and marked-up manuscript are given below.

**Comment 1:**
"The manuscript does an exceptionally poor job of describing what the current understanding is - both observationally and in terms of models. And the specific problem to be addressed is also poorly articulated."

Response:
Section 1 for the "introduction" was rewritten and section 6 for "summary and discussion" were newly added to clarify the aims and the results of this work.

**Comment 2:**
"An attempt at clarifying the main problem is added in lines 23-27. This refers to the balloon-measured E-fields, Ba release experiments, and radar studies from the 1970s. As a reader, I am simply not convinced that those studies have led the community to the idea that the poleward motion of auroras travelling at different rates than the equatorward drifts is at all an unanswered question. Surely the author can produce references that have been published over the past 40 years that may have explained these observations?"

Response:
Please refer to the following sentences added in Lines 31-38 of the section 1 (Introduction): "To account for the difference in propagation directions, it was suggested that the primary sources of auroral particles are in the magnetospheric plasmas and they developed in terms of propagation of rarefaction wave in the tail [Chao et al., 1971; Liu et al., 2012], tailward regression/braking of the fast earthward flows referred to as BBFs [Shiokawa et al., 1997; Haerendel, 2015], and onset instability of inner plasma sheet pressure [Nishimura et al., 2010]. It is suggested that substorm and poleward expansion of auroras were initiated and amplified at the substorm onset by the BBFs arriving at the inner boundary of plasma sheet from the tail [Kepko et al., 2004; Angelopoulos et al., 2008; Machida et al., 2009]."

**Comment 3:**

"There was an addition of two sentences describing the current understanding on poleward propagation of the substorm auroras. Unfortunately, the explanation is too brief, with no references given and is substantially incorrect. A tailward moving reconnection site is not needed for poleward motion in the reconnection-based model -- simply having reconnection progress to the lobes is enough."

Response:

Please refer to our above response in Comment 2 for our understanding on poleward propagation of the substorm auroras. Reconnection progress is not included in our explanation, but it may be the subject of another paper, Lines 257-261.

**Comment 4:**

"There is a very alarming trend in this paper where it seems that every citation of facts relating to observations or causative mechanisms are to the author's own previous publications -- none of which I am familiar with. There certainly are other studies on the aurora and Pi2s, etc. that are relevant to the current topic. It is completely inappropriate not to cite these many other papers."

Response:

I tried to include as many relevant works as possible. In the previous version of the manuscript, I perhaps too narrowly focused on my own studies to explain the role of low latitude Pi2 pulsations in the substorms. In this revised version, I have remained only more essential references to my work.

**Comment 5:**

"The new section #2 does not provide any confidence at all in the assertions being made. Use of the T89 Kp 4 model to map results is notoriously untrustworthy. The results may or may not be correct -- which doesn't really tell us anything. There are also many facts definitively stated about Pi2s that by themselves warrant much more observational support than is given here. Inclusion of equation (1) is unnecessary since its context is completely lost on the reader. The conclusions drawn from this equation in lines 71-73 require far more convincing explanation than is given. The remainder of this section (on ballooning) is unclear. Is this a new idea? Existing idea? Is it based on models, theory, observations?

The final sentence of this section is typical of the confusing and unsupported claims in the paper; "The convection surge occurred once in the initial pulse of Pi2 pulsation but is not repeated in the following pulses." This statement seems important, but it is incomprehensible to me. How does the author know this? Is this referring to the Rubtsov study? What was that study? Did they demonstrate that this was a universal finding for substorms? How does this relate to the rest of the current paper? None of this is made clear."

Response:

(1) Please refer to the following sentences in Lines 64-71, "We can postulate the onset scenario that Bursty Bulk Flows (BBFs) reaching the geosynchronous orbit activated preonset auroras in lower latitudes by the transmission of electric fields from the dipolarization front (DF) embedded in the initial pulse of the BBF [Runov et al., 2011]. These transient electric fields were observed by the geosynchronous satellite as the convection enhancement of the plasma sheet electrons due to local breakdown of the last open trajectories of plasma sheet electrons [Thomsen et al., 2002]. The convection enhancement occurred in all-sky image coincident with the onset of bead-like rippling that leads to the breakups at equatorward latitudes [Saka et al., 2014]."

We suppose that the use of T89 model for field line mapping may be acceptable for the present purpose.

(2) Equation (1) and sentences relating to the ballooning instability were eliminated.

(3) Please refer to the following sentences in Lines 71-79, "In the Pi2 pulses following the initial pulse, an auroral surge was observed in all-sky images between $66^{\circ}$N to $74^{\circ}$N in geomagnetic latitudes referred to as Poleward Boundary Aurora Surge (PBAS) [Saka et al., 2012]. They propagated eastward or westward at the poleward boundary of the auroral zone and were interpreted as an auroral manifestation of flow bifurcation of BBFs. In this onset scenario, the field line dipolarization finished in the initial pulse of the Pi2 pulsations, increasing field line inclination in a step-like manner for Goes5, and generating transient pulses for Goes6. The convection surge occurred once in the initial pulse of BBFs (DF) but is not repeated in the following pulses in the BBF train. This correlation suggests that auroral breakup may not repeat in the Pi2 wave packet but occurred at its initial pulse."

**Comment 6:**

"Section #3 has a paragraph added, but it is also unclear. In line 83 it is stated that "It is reasonable to assume..." Why is it reasonable to assume this? Where does the surge come, from? Why would the high latitude end not expand as stated? Why would the flows be confined as stated? There are simply too many unsupported statements here. And then there is only a single reference to the author's own 2014 publication."

Response:

Please refer to the following sentences in Lines 64-71 in section 2 for the explanation of the convection surge; its location and onset timing. "The convection surge came from the dipolarization front at the leading edge of BBFs referred to as DF [Runov et al., 2011]. They often appeared at the geosynchronous altitudes as convection enhancement of plasma sheet electrons due to local breakdown of the last open trajectories of plasma sheet electrons [Thomsen et al., 2002]. It appeared in all-sky image coincident with the onset of bead-like rippling that leads to the breakups at equatorward latitudes [Saka et l., 2014]."

**Comment 7:**

"Section 4 is all bold-faced implying that there are major modifications to the manuscript, but much of it seems identical. I am not sure what has been changed there. Also, equation (6) is from Kelley's text book. Please indicate where in the textbook it can be found. This section also still reads more like facts are being disclosed rather than like an idea is being proposed. I don't know what's known or what's being proposed from this."

Response:

(1) Steady state flow came from (2.34) in [Kelley, 1989]. Eliminating the second term in the left-hand side of (2.34), we have equation (6) for the parallel flows. It can be applied for both collisional and collisionless cases.

(2) We proposed that compressional ionosphere created the ion outflows and inverted-V type electron accelerations through the excitation of ion acoustic wave in the ionosphere. This idea is new because the ionosphere has been previously considered an inhomogeneous but incompressive medium.

**Comment 8:**

"Section 5 is also all bold-faced making it difficult to see what has actually been changed."

Response:

No change was made in this section.

**Comment 9:**

"The conclusion is still not supported by the body of the paper. I remain unconvinced that "this scenario, analogous to traffic flow of cars on the crowded roads, partly explains the discrepant time history of the auroras which is often described as the auroras expand opposite to that of plasma drift in the ionosphere." The author claims this, but there are many unsupported components that go into the scenario. A rather disturbing aspect of the paper is that it really still does not articulate what the problem is. Citing a few papers from the 1970's is not adequate here -- note that BBFs, streamers, and their relationship, etc were not known to those authors.

Also, the manuscript is completely devoid of any meaningful description of what the auroral observations really show during substorms. In re-reading this paper many times, my guess (and I have to stress here it is just a guess) is that the author is trying to explain the following observational scenario; a) a BBF-associated streamer propagates equatorward, b) as it interacts with the equatorial region, the presumed density accumulations described in the proposed scenario lead to poleward propagation of the auroras, c) this explains the hypothesized substorm sequence described by Nishimura et al.

If this is the intent of the currently proposed scenario, it is not at all clear from what is written. In addition, it is almost certainly incorrect. The types of events showing rapid poleward motion in response to streamers (the so-called contact breakups discovered by Oguti in the 1970s) tend to be explosive in nature, not just like a sand pile building up or tail lights propagating backward. (Note that this is also a major problem with the flow-braking model.) In addition, as I stated above and in my previous review, the scenario proposed here (as well as the Nishimura one) does not adequately describe things like plasmoid releases that are known to occur with substorms.

Once again, I suggest that the author rewrite the manuscript in a manner that the reader can understand what problem it is that is actually trying to be addressed. I feel that there are interesting and possibly important issues raised by the proposed scenario, but the presentation and conclusions are still very misguided. Below is a suggested outline of topics to cover."

Response:

(1) This report described the auroral expansion in terms of: (1) the BBF triggering the convection surge by electric fields in DF; (2) projections of these electric fields to the ionosphere; (3) creation of the compressibility in the ionosphere by the electric field drifts; (4) excitation of ion acoustic wave by the compression; (5) generation of parallel electric fields by ion acoustic wave; (6) nonlinear evolution of the compressibility for the poleward expansion of auroras. The auroral expansion was described in this acoustic regime which we believe a new scenario not proposed previously.

(2) Characteristic speed of the auroral formation was reported to be 5-8 km/s independent of the spatial scale of the auroras [Oguti, Metamorphoses of aurora, Memoirs of National Institute of Polar Research: series A, aeronomy, 12, 1-101, 1975]. If we can use this velocity as a speed of poleward expansion, the total electric fields of the order of 300 mV/m may be required in the auroral ionosphere. The electric fields from DF (5 mV/m at B=30 nT) create incident electric fields of the order of 200 mV/m by the projections into the auroral ionosphere (B=50000 nT). If the Alfven conductance was larger than the height integrated Pedersen conductance, total electric fields (sum of incident and reflected electric fields) may increase to 300 mV/m. These electric fields produce poleward expansion velocity of the order of 6 km/s. This is consistent with Oguti's results.

(3) Please refer to Lines 257-261. We stated that "We note that poleward expansion as described here is an auroral event occurring in the initial pulse of Pi2 pulsations. In the succeeding pulses in the Pi2 wave trains, auroras are composed of poleward surge propagating at the poleward boundary of auroral zone (PBAS) [Saka et al., 2012]. We suppose that PBASs may be directly correlated to the reconnection processes inherent in the plasma sheet. This topic will be explored in another paper."

**Comment 10:**
"1) Introduction a) Describe phenomenologically what a substorm is both from an auroral point of view and from a tail point of view. This should not just cite the author's work from a few years ago or papers from 40-60 years ago. It should also be extensive enough to demonstrate to the reader that the author has a good grasp of what the current understanding is. The present manuscript does not convey this at all. This should be more than a quote about the importance of aurora from Oguti or a reference to Akasou's 1960's papers.

b) Describe what models are currently accepted to describe these observations. (Yes they exist and there is more than one.) Again don't just cite the author's work from a few years ago or papers from 40-60 years ago.

b) Have a section on a statement of the specific unresolved issue to be targeted here. What is the current state of understanding on the poleward/equatorward issue in particular? What are the successes/deficiencies in these ideas? Why does it make sense to dismiss them? Or to modify them?

c) briefly describe what the current paper will do to resolve this problem, with a short outline of the sections to follow and how they flow."

Response:
Please refer to section 1 for the introduction. It was rewritten to include above comment. The BBF triggering has been accepted substorm model in the literature. I emphasized in the manuscript that auroral ionosphere becomes more active in this context.

**Comment 11:**
"2) Proposed new model type of section. Describe the scenario and how it can lead to poleward propagation.

a) Sections on the scenario -- these are largely written already, but need to be drastically cleaned up to read more coherently. (With fewer definitive assertions and more suggestions.)"

Response:
Please refer to sections 2, 3, 4, and 5 for proposed scenario. We suggest that plasma compression in the ionosphere implemented the ionosphere active. The active ionosphere includes a nonlinear evolution of the compressed ionosphere, field-aligned currents to satisfy the quasi-neutrality of the ionosphere, and parallel potentials associated with the excitation of an ion acoustic wave. We studied how the active ionosphere created auroral expansion.

**Comment 12:**

"3) Discussion-like sections a) Describe what types of poleward propagation it can address? Note that it is totally unbelievable that this scenario can address all facets of poleward motion. Why? Because at the extreme end, the poleward propagating part of the bulge eventually forms a double-oval like configuration that must engage extremely large regions of the magnetotail. Unless the density-accumulation concept proposed here can engage most of the nightside magnetosphere, it seems insufficient.

b) Can the scenario yield explosive poleward motion? Why, why not? Etc.. (Note that tailward-propagating tail-lights doesn't seem explosive.)

c) How does this scenario relate to other ideas and how does it explain all of the observations that were described in the first section? E.g. flow-braking? What has been neglected for the current scenario to ignore flow-braking? Etc.."

Response:

(a) Please refer to Lines 257-261; "we note that poleward expansion as described here is an auroral event occurring in the initial pulse of Pi2 pulsations. In the succeeding pulses in the Pi2 wave trains, auroras are composed of poleward surge propagating at the poleward boundary of auroral zone (PBAS) [Saka et al., 2012]. We suppose that PBASs may be directly correlated to the reconnection processes inherent in the plasma sheet. This topic will be explored in another paper."

(b) Characteristic speed of the auroral formation was reported to be 5-8 km/s independent of the spatial scale of the auroras [Oguti, Metamorphoses of aurora, Memoirs of National Institute of Polar Research: series A, aeronomy, 12, 1-101, 1975]. If we can use this velocity as a speed of poleward expansion, the total electric fields of the order of 300 mV/m may be required in the auroral ionosphere. The electric fields from DF (5 mV/m at B=30 nT) create incident electric fields of the order of 200 mV/m by the projections into the auroral ionosphere (B=50000 nT). If the Alfven conductance was larger than the height integrated Pedersen conductance, total electric fields (sum of incident and reflected electric fields) may increase to 300 mV/m. These electric fields produce poleward expansion velocity of the order of 6 km/s. This is consistent with Oguti's results.

(c) The BBF triggering has been accepted substorm model in the literature. I emphasized in the manuscript that auroral ionosphere becomes active in this context. The active ionosphere was applied to describe the auroral expansion.

**Comment 13:**

"3) Conclusion/discussion on where the scenario is most likely to fit into the overall picture. My feeling is that it might provide some explanation of the poleward motion during situations where streamers reach the equatorward part of the oval and possibly during some early initial phase of the contact breakup type events. (Note that the first disturbance is not a substorm at all and the second may only relate to the early phases of a complete substorm.) I rather suspect that the scenario is extremely unlikely to be able to explain the entire typical substorm sequence. A competent description of "what a substorm is" in the first section will, almost certainty, lead to the later part of this conclusion. Without that first section, conclusions like the one in the present manuscript are simply not supported."

Response:

We described the auroral expansion in terms of acoustic regime because the ionosphere becomes active by the compression. If the field line thinning developed enough in the growth phase, intense electric fields in DF reached lowest latitudes to initiate active ionosphere. If the thinning was not enough, weak electric fields initiate weak or no activities at intermediate latitudes.

[revised manuscript text omitted]

Figure 1

[Figure]

Figure 2

[Figure]

Figure 3

[Figure]

Figure 4

(A)

[Figure]

(B)

[Figure]

Figure 5